# Seroprevalence and associated factors of HIV and Hepatitis C in Brazilian high-security prisons: A state-wide epidemiological study

Lirane Elize Defante Ferreto[1,2], Stephanny Guedes[3], Fernando Braz Pauli[2], Samyra Soligo Rovani[2], Franciele Aní Caovilla Follador[2], Ana Paula Vieira[2], Renata Himovski Torres[4], Harnoldo Colares Coelho[5], Guilherme Welter Wendt[1]*

1 Western Paraná State University, Health Sciences Center, Faculty of Medicine, Public Health Lab, Francisco Beltrão, Brazil, 2 Western Paraná State University, Health Sciences Center, Postgraduate Program in Applied Health Sciences, Francisco Beltrão, Brazil, 3 Western Paraná State University, Faculty of Medicine, Cascavel, Brazil, 4 Division of Public Security and Penitentiary Administration, Penitentiary Department, Curitiba, Brazil, 5 São Paulo State University, Department of Pharmacy, Ribeirão Preto, Brazil

☯ These authors contributed equally to this work.
* guilhermewwendt@gmail.com

**Data Availability Statement:** Data cannot be shared publicly because of the Resolution NHC

## Abstract

The prevalence of hepatitis C virus (HCV) and the acquired immunodeficiency virus (HIV) is much higher in prisons than in community settings. Some explanatory factors for this burden include putative aspects of the prison environment, such as unprotected sexual relations and sexual violence, use of injectable drugs and syringe sharing. Nonetheless, efforts in better understanding the dynamics of both HCV and HIV are scarce in developing countries such as Brazil, which poses a risk not only to the inmates but to the community as well. In this investigation, we sought to determine the seroprevalence and sociodemographic and behavioral risk factors associated with HIV and anti-HCV antibodies among men detained at high-security institutions. This is an epidemiological, proportionally stratified observational study including 1,132 inmates aged 18 to 79 years-old ($M_{age}$ = 32.58±10.18) from eleven high-security prisons located in the State of Paraná, Brazil. We found that HIV and anti-HCV prevalence were 1.6% (95% CI: 1.0–2.5) and 2.7% (95% CI: 1.0–2.5), respectively. Risk factors associated with HIV included not receiving intimate visits (OR = 8.80, 95% CI: 1.15–66.88), already having another sexually transmitted infection (OR = 3.89, 95% CI: 1.47–10.29), and reporting attendance in HIV preventive campaigns (OR = 4.24, 95% CI: 1.58–11.36). Moreover, anti-HCV seroprevalence was associated with higher age (OR = 4.03, 95% CI: 1.61–10.07), criminal recidivism (OR = 2.58, 95% CI 1.02–6.52), and the use of injectable drugs (OR = 7.32, 95% CI 3.36–15.92). Although prisons might increase the risk for acquiring and transmitting HIV and HCV, the adoption of permanent epidemiological surveillance programs could help reducing the circulation of viruses, involving strategies focusing on screening, treating, and preventing infections to assure proper prisoner health. Moreover, these policies need to take place inside and outside the prison environment to offer continued assistance to former prisoners once they leave the institution.

number 466 (http://conselho.saude.gov.br/resolucoes/2012/Reso466.pdf), items "II.25", "III 1 q", and "IV.3 e" states that researchers might use and share the material and data obtained in the research exclusively according to the consent of the participant and/or institutions and sharing these details must be approved beforehand. Data are available from the Western Paraná State University Institutional Ethics Committee (contact via cep.prppg@unioeste.br) for researchers who meet the criteria for access to confidential data.

**Funding:** This study has been funded by the Brazilian Ministry of Health [Ministério da Saúde - BR; grant 797322/2013].

**Competing interests:** The authors have declared that no competing interests exist.

## Introduction

Viruses causing both hepatitis C (HCV) and the acquired immunodeficiency virus (HIV) are of critical concern in developed and developing countries, including Brazil [1, 2]. Between 1999 to 2018, 359,673 HCV cases were reported in Brazil; during this same period, detection rates in Southern regions were higher compared to other regions, with 26.8 cases per 100,000 inhabitants [3]. In relation to HIV, cumulative cases from 1980 to 2020 indicate a total of 1,011,617 diagnosis [4].

Despite the high rates of HCV and HIV infection in the community, there are populations with even greater vulnerability, such as prisoners [5, 6]. Indeed, prisoners are 48 to 69 times more likely to be infected with HCV and 7 to 12 times more likely to be infected with HIV when compared to the general population [7]. Moreover, there are around 10 million incarcerated individuals across de world, with 3.8% and 15.1% infected with HIV and HCV, respectively [8]. However, substantial regional variation in these estimates exists. For instance, HCV prevalence in Eastern Europe and central Asia region is around 20.2%, while numbers decline in Western Europe (15.5%), and in North (15.3%) and Latin America (4.7%); additionally, HIV prevalence among prisoners is also higher in East and South Africa (15%) in comparison to the 5% reported in Central Asia, Eastern and Western Europe [8].

Some scholars argue that HIV and HCV prevalence are higher in prisoners due to widespread criminalization of addiction, leading to conviction and detention of drug users [8, 9]. Despite the specific risks underpinning infection among drug users, the prison environment itself accentuates the odds of acquiring infectious diseases. This occurs as risky behaviors–including injecting drug use (IDU), needle sharing, tattoos and unprotected sexual practices–tend to increase when individuals are incarcerated [1]. HIV and HCV infection also varies between subgroups of inmates, especially among men who have sex with men (MSM), sex workers (SWs), IDU, and transgender women (TGW) [9–11]. The consequences of incarceration and infection by HIV and HCV are vast and well-documented, having a cascading effect on multiple levels. For individuals, synergistic deficits increase the susceptibility for disease progression, multimorbidity, psychiatric suffering, and death [12, 13]. For prison health and management, the duty of properly caring for serious physical and mental needs, coupled with efficient integration with external health services, are certainly challenging [14]. Lastly, societal implications must be considered, including public health concerns as former prisoners might not seek specialized and early care due to untreated conditions, notwithstanding the higher chances of recidivism and difficulties in resocialization commonly seen [13, 15].

HCV is about ten times more transmittable by percutaneous exposure and has an even greater association with IDU than HIV [16]. Moreover, IDU are six times more likely of contracting HIV while in prison [17, 18]. Thus, the absence of interventions and access to treatment for both sexually transmitted infections (STIs) and addition might maintain high levels of syringe sharing, which then raises transmission rates and risks for reincarceration [17].

Despite the syndemic of STIs, data on HIV and viral hepatitis in prison populations are still insufficient and or poorly reported, especially in low-income nations [19, 20]. To overcome the limitations regarding prisoners' health and the limited resources for prevention and treatment [9, 18, 21], this investigation sought to explore the seroprevalence and associated risk factors for HIV and HCV infection in male inmates. We anticipated that IDU, previous STIs, and unprotected sexual behaviors would be associated with both HIV and HCV [1].

## Methods

### Study design and setting

This is a state-wide epidemiological and observational study conducted in 11 maximum security prisons for males, chosen out of a total of 23 institutions with around 19,000 total inmates

from May 2015 to December 2016. The institutions are spread across the Paraná state (Brazil), covering small (Francisco Beltrão; FBE), medium (Londrina; LON), and large cities (Curitiba metropolitan region; CWB), thus representative of the large population of prisoners. In FBE, data were collected at the State Penitentiary; in LON, data was collected at the LON State Penitentiary I, LON State Penitentiary II, and in the Center for Social Reintegration; in the CWB metropolitan area, data were gathered from Piraquara State Penitentiary I and II, the Central State Penitentiary, Curitiba Forensic Unit, and from the Detention Centers in São José dos Pinhais, Piraquara, and Curitiba. The criteria for selecting those cities were based on the fact of having maximum security prisons that provide daily supervision within the prison limits, and for representing different demographic scenarios. Proportional stratified sampling was performed using each prison as a randomization unit.

## Participants

The population included 8,142 inmates, and the sample size (n) involved 1,132 inmates ($M_{age}$ = 32.58±10.18; range: 18 to 79 years). Participants were serving sentences of an average of 12.64 years ($SD$ = 13.55), ranging from less than one year to 99 years. Sample size was calculated based on an expected anti-HIV seroprevalence of 50%±1% [22, 23] with a power of 80% and $\alpha$ = .03, which yielded the necessity of 954 inmates.

## Procedures

Participation in the study was voluntary and no compensation was provided. Individuals were informed about their option to end their participation in the study at any time without consequences, thus offering to everyone the opportunity to be screened for HIV and anti-HCV. Ethical clearance was obtained from the Western Paraná State Research Ethics Committee (n. 33349314.9.0000.0107). On the day of data collection, prisoners were ranked numerically in ascending order according to lists provided by each prison unit. A list of random numbers was generated using Microsoft Excel® and participants were randomly chosen. To be included, participants had to be in state's custody, be ≥18 years and able to consent for themselves in taking part in the research. Written informed consents were obtained from all individuals. Those who agreed to share the results from the serological testing with the institutions medical personnel had their records updated and, if necessary, guidance was offered to health staff.

An interview was conducted to gather information of potential risk factors for HIV and HCV infection (i.e., sexual behavior, drug usage, among others [Table 1]) in addition to socio-demographic variables (i.e., age, education, among others [Table 1]). Subsequently, participants were submitted to blood collection via venous puncture. Blood analyses were performed at Biocenter Lab (Foz do Iguaçu, Brazil) involving fourth-generation ARCHITECT HIV Ag/Ab combo assay® (Abbott Diagnostics, Wiesbaden, Germany). HIV reagent results were confirmed with Geenius™ HIV 1/2 confirmatory assay and Western Blot (Biorad®, Santo Amaro, Brazil). Inconclusive results were tested twice and received reagent, non-reactive, or inconclusive diagnosis. ARCHITECT Anti-HCV high throughput chemiluminescent microparticle immunoassay was used to investigate HCV seroprevalence (Architect® Anti-HCV Assay, Germany).

## Data analysis

Data were entered and analyzed using the Statistical Package for the Social Science (SPSS), version 24.0 (SPSS Inc., Chicago, IL, USA). Descriptive and inferential statistics were used to determine the prevalence of HIV and anti-HCV as well as to describe the study's sample. Multivariate modelling was used to obtain raw and adjusted odds ratios (OR) of factors associated

**Table 1. Demographic information from 1,132 inmates from 11 high-security prisons in the State of Paraná, Brazil (n = 1,132).**

| | n | % |
|---|---|---|
| **Municipality** | | |
| Francisco Beltrão | 119 | 10.51 |
| Londrina | 276 | 24.38 |
| Curitiba | 737 | 65.10 |
| **Age (years)** | | |
| Between 18 and 30 | 582 | 51.41 |
| More than 30 | 550 | 48.58 |
| **Ethnicity** | | |
| White | 447 | 39.48 |
| Brown | 477 | 42.13 |
| Asian | 23 | 2.03 |
| Black | 156 | 13.78 |
| Indigenous | 29 | 2.56 |
| **Marital status** | | |
| Single | 604 | 53.35 |
| With someone | 528 | 46.64 |
| **Has children** | | |
| Yes | 750 | 66.25 |
| No | 382 | 33.74 |
| **Education** | | |
| Incomplete primary education | 1082 | 95.59 |
| Complete primary education | 50 | 4.41 |
| **Recidivism** | | |
| None or one | 725 | 64.04 |
| Two or more | 407 | 35.95 |
| **Sentence time** | | |
| Up to 10 years | 1090 | 96.30 |
| More than 10 years | 42 | 3.7 |
| **Sexually transmitted infection** | | |
| Yes | 172 | 15.20 |
| No | 960 | 84.80 |
| **Having tattoo** | | |
| Yes | 808 | 71.37 |
| No | 324 | 28.62 |
| **Having piercings** | | |
| Yes | 420 | 37.10 |
| No | 712 | 62.90 |
| **Sharing objects** | | |
| Yes | 839 | 74.11 |
| No | 293 | 25.88 |
| **Knowledge on HIV** | | |
| Yes | 917 | 81.00 |
| No | 215 | 19.00 |
| **Knowledge on HCV** | | |
| Yes | 444 | 39.22 |
| No | 688 | 60.77 |

*(Continued)*

**Table 1.** (Continued)

|  | n | % |
|---|---|---|
| **Attending preventive campaigns** |  |  |
| Yes | 90 | 7.95 |
| No | 1042 | 92.05 |
| **Blood transfusion** |  |  |
| Yes | 156 | 13.8 |
| No | 976 | 86.20 |
| **Having sexual relationships with drug users** |  |  |
| Yes | 670 | 59.20 |
| No/Do not know | 462 | 40.80 |
| **Being a drug user** |  |  |
| Yes | 850 | 75.00 |
| No | 282 | 25.00 |
| **Alcohol use** |  |  |
| Yes | 1062 | 93.81 |
| No | 70 | 6.18 |
| **Being an inject drug user** |  |  |
| Yes | 97 | 8.56 |
| No | 1035 | 91.43 |
| **Self-reported sexual orientation** |  |  |
| Heterosexual | 1034 | 91.34 |
| Other | 98 | 8.65 |
| **Having homosexual sex** |  |  |
| Yes | 66 | 5.83 |
| No | 1066 | 94.17 |
| **Receiving intimate visits** |  |  |
| Yes | 372 | 32.86 |
| No | 760 | 67.13 |
| **Receiving/having condoms in prison** |  |  |
| Yes | 564 | 49.82 |
| No/Do not know | 568 | 50.18 |

with positive HIV and anti-HVC results. Following past research on the matter [6, 20], adjusted OR with 95% confidence intervals (CI) were calculated in the multivariate logistic analyses to reduce bias regarding unequal sizes in the comparison groups.

## Results

Table 1 provides a general overview of the demographic information about the studied samples. Notably, participants had low educational levels, with elevated proportion of drug users. The number of inmates reporting having sexual intercourse with drug users or not knowing whether their sexual partners were drug users was also noteworthy.

Table 2 presents the overall seroprevalence of anti-HVC antibodies and HIV in the total sample and in relation to a series of demographic and behavioral variables. Remarkably, MSM, aged over 30, with higher recidivism, without piercings and reporting IDU showed statistically higher seroprevalence of anti-HCV. On the other hand, prevalence of HIV was higher in individuals who already had STIs and in those that reported having participated in preventive campaigns against STIs, and in men who did not receive intimate visits.

**Table 2. Prevalence of anti-HCV and HIV in 1,132 from 11 high-security prisons in the State of Paraná, Brazil (n = 1,132).**

| | anti-HCV | | p | HIV | | p |
|---|---|---|---|---|---|---|
| | Frequency of positive results | Prevalence (95% CI) | | Frequency of positive results | Prevalence (95% CI) | |
| Overall | 30 | 2.7 (1.9–3.8) | | 18 | 1.6 (1.0–2.5) | |
| **Municipality** | | | .12 | | | .26 |
| Francisco Beltrão | 1 | 0.8 (0.1–4.0) | | 0 | nc | |
| Londrina | 6 | 2.2 (0.9–4.4) | | 5 | 1.8 (0.6–4.3) | |
| Curitiba | 23 | 3.1 (2.2–5.8) | | 13 | 1.8 (1.0–3.0) | |
| **Age (years)** | | | **.001** | | | .91 |
| Between 18 and 30 | 6 | **1.0 (0.1–2.0)** | | 10 | 1.7 (0.8–3.2) | |
| More than 30 | 24 | **4.4 (2.9–6.4)** | | 8 | 1.5 (0.6–2.9) | |
| **Education** | | | .86 | | | .40 |
| Incomplete primary education | 12 | 2.7 (1.6–4.5) | | 7 | 1.6 (0.6–3.2) | |
| Complete primary education | 18 | 2.6 (1.5–4.2) | | 11 | 1.6 (0.8–2.8) | |
| **Recidivism** | | | **.01** | | | .09 |
| None or one | 6 | **1.2 (0.5–2.5)** | | 4 | 0.8 (0.2–2.1) | |
| Two or more | 24 | **3.8 (2.6–5.7)** | | 14 | 2.2 (1.3–3.8) | |
| **Sentence time** | | | .35 | | | .31 |
| Up to 10 years | 13 | 2.1 (1.2–3.7) | | 7 | 1.2 (0.5–2.4) | |
| More than 10 years | 17 | 3.2 (2.0–5.1) | | 11 | 2.1 (1.1–3.7) | |
| **Sexually transmitted infection** | | | .18 | | | **.003** |
| Yes | 12 | 3.8 (2.1–6.7) | | 11 | **3.5 (1.9–6.3)** | |
| No | 17 | 2.2 (1.4–3.5) | | 7 | **0.9 (0.3–1.8)** | |
| **Having tattoo** | | | .86 | | | .48 |
| Yes | 9 | 2.8 (1.4–5.3) | | 7 | 2.2 (0.9–4.5) | |
| No | 21 | 2.6 (1.7–4.0) | | 11 | 1.4 (0.7–2.5) | |
| **Having piercings** | | | **.03** | | | .37 |
| Yes | 5 | **1.2 (0.4–2.8)** | | 9 | 2.1 (1.1–4.1) | |
| No | 25 | **3.5 (2.4–5.1)** | | 9 | 1.3 (0.6–2.4) | |
| **Sharing objects** | | | .46 | | | .24 |
| Yes | 10 | 3.4 (1.8–6.3) | | 16 | 1.9 (1.2–3.1) | |
| No | 20 | 2.4 (1.5–3.7) | | 2 | 0.7 (0.1–2.6) | |
| **Knowledge on HIV** | | | .92 | | | .58 |
| Yes | 25 | 2.7 (1.8–4.0) | | 16 | 1.7 (1.1–2.8) | |
| No | 5 | 2.3 (0.8–5.5) | | 2 | 0.9 (0.1–3.6) | |
| **Knowledge on HCV** | | | .30 | | | .78 |
| Yes | 15 | 3.4 (2.0–5.5) | | 6 | 1.4 (0.5–3.0) | |
| No | 15 | 2.2 (1.3–3.6) | | 12 | 1.7 (0.9–3.1) | |
| **Attending preventive campaigns** | | | .95 | | | **.001** |
| Yes | 3 | 3.1 (0.9–8.0) | | 6 | **5.5 (2.5–11.1)** | |
| No | 26 | 2.6 (1.8–3.8) | | 11 | **1.1 (0.6–2.0)** | |
| **Blood transfusion** | | | .22 | | | .99 |
| Yes | 7 | 4.4 (2.0–9.0) | | 2 | 1.3 (0.1–4.8) | |
| No | 23 | 2.4 (1.6–3.5) | | 16 | 1.6 (0.9–2.7) | |
| **Having sexual relationships with drug users** | | | .92 | | | .37 |
| Yes | 18 | 2.7 (1.7–4.2) | | 13 | 1.9 (1.1–3.3) | |
| No/Do not know | 10 | 2.6 (1.4–4.5) | | 3 | 1.1 (0.3–2.6) | |
| **Being a drug user** | | | .09 | | | .27 |

*(Continued)*

**Table 2.** (Continued)

| | anti-HCV | | p | HIV | | p |
|---|---|---|---|---|---|---|
| | **Frequency of positive results** | **Prevalence (95% CI)** | | **Frequency of positive results** | **Prevalence (95% CI)** | |
| Yes | 27 | 3.2 (2.2–4.6) | | 16 | 1.9 (1.1–3.1) | |
| No | 3 | 1.1 (0.2–3.2) | | 2 | 0.7 (0.1–2.7) | |
| **Alcohol use** | | | .20 | | | .91 |
| Yes | 26 | 2.4 (1.7–3.6) | | 17 | 1.6 (1.0–2.6) | |
| No | 4 | 5.7 (1.8–14.2) | | 1 | 1.4 (0.1–12.4) | |
| **Being an inject drug user** | | | .001 | | | .41 |
| Yes | 13 | **13.4 (7.9–21.8)** | | 3 | 3.1 (0.7–9.1) | |
| No | 17 | **1.6 (1.0–2.6)** | | 15 | 1.4 (0.8–2.4) | |
| **Self-reported sexual orientation** | | | .95 | | | .42 |
| Heterosexual | 28 | 2.7 (1.9–3.9) | | 15 | 1.5 (0.9–2.4) | |
| Other | 2 | 2.0 (0.1–7.6) | | 3 | 3.1 (0.7–9.0) | |
| **Having homosexual sex** | | | .03 | | | .65 |
| Yes | 5 | **7.6 (2.9–6.9)** | | 2 | 3.0 (0.2–1.0) | |
| No | 25 | **2.3 (1.6–3.5)** | | 16 | 1.5 (0.9–2.4) | |
| **Receiving intimate visits** | | | .60 | | | .02 |
| Yes | 8 | 2.2 (1.0–4.3) | | 1 | **0.3 (0.1–1.7)** | |
| No | 22 | 2.9 (1.9–4.4) | | 17 | **2.2 (1.4–3.6)** | |
| **Receiving/having condoms in prison** | | | .83 | | | .80 |
| Yes | 16 | 2.8 (1.7–4.6) | | 10 | 1.8 (0.9–3.3) | |
| No/Do not know | 9 | 2.5 (1.4–4.1) | | 5 | 1.4 (0.7–2.8) | |

CI: Confidence Intervals; NC: non computed; In bold, significant differences between rows.

Tables 3 and 4 present factors associated with anti-HCV serological status and HIV infection among 1,132 high-security inmates, respectively. Age, recidivism, and IDU were significantly associated with the presence of anti-HCV antibodies (Table 3), while not receiving

**Table 3. Factors associated with anti-HCV antibodies seroprevalence in 1,132 high-security inmates.**

| | OR$_{raw}$ (95% CI) | p | OR$_{adjusted}$ (95% CI) | p |
|---|---|---|---|---|
| **Age** | | | | |
| Between 18–30 | 1 | | | |
| > 30 | 4.38 (1.77–10.80) | **.001** | 4.03 (1.61–10.07) | **.003** |
| **Recidivism** | | | | |
| Yes | 3.33 (1.35–8.22) | **.009** | 2.58 (1.02–6.52) | **.04** |
| No | 1 | | | |
| **Has piercing** | | | | |
| Yes | 1 | | — | — |
| No | 3.02 (1.15–7.95) | **.02** | | |
| **Injecting drug user** | | | | |
| Yes | 9.27 (4.35–19.73) | **< .001** | 7.32 (3.36–15.92) | **< .001** |
| No | 1 | | | |
| **Homosexual relations** | | | | |
| Yes | 3.41 (1.26–9.22) | **.016** | — | — |
| No | 1 | | | |

CI: Confidence Intervals. OR: Odds ratios; In bold, significant differences between rows.

**Table 4. Factors associated with HIV infection in 1,132 high-security inmates.**

| | OR$_{raw}$ (95% CI) | p | OR$_{adjusted}$ (95% CI) | p |
|---|---|---|---|---|
| **History of sexually transmitted infection** | | | | |
| Yes | 4.23 (1.62–11.00) | **.003** | 3.89 (1.47–10.29) | **.006** |
| No/Do not know | 1 | | 1 | |
| **Attended preventive campaigns for HIV/HCV** | | | | |
| Yes | 5.22 (1.99–13.73) | **.001** | 4.24 (1.58–11.36) | **.004** |
| No | 1 | | 1 | |
| **Receives intimate visit** | | | | |
| Yes | 1 | | 1 | |
| No | 8.49 (1.12–64.03) | **.038** | 8.80 (1.15–66.88) | **.036** |

CI: Confidence Intervals. OR: Odds ratios; In bold, significant differences between rows.

intimate visits, attending STIs preventive campaigns and possessing other STIs were linked with higher odds of having HIV (Table 4). Independently from the other variables, individuals who reported injecting drugs had seven times more chances for presenting with anti-HCV antibodies in comparison to those that did not exhibit this behavior.

Multivariate analyses for HIV outcome maintained the raw OR in the adjusted, final model. In summary, when compared to individuals who received intimate visits, those who reported not receiving them had nine times more chances of HIV infection. Those who reported having STIs and those who participated in preventive campaigns against them were also more likely to have a positive HIV status.

## Discussion

In this investigation, the goal was to explore the sociodemographic and behavioral risk factors associated with HIV and anti-HCV seroprevalence in men detained at high-security institutions, expanding on past research conducted in other Brazilian states that used mixed samples of inmates [20, 24]. Data describes a profile of vulnerability among participants, in which low education and high drug usage was reported. These factors could partly explain the burden of infectious diseases and other physical and mental suffering among Brazilian inmates [19] and are quite different from studies conducted in other regions of the country. For instance, in the current research, the percentage of IDU was way higher than the one reported by Sgarbi and colleagues (0%) and by Puga and collaborators (5.56%) in the Brazilian state of Mato Grosso, while lower education was much more frequent in our investigation in comparison to these investigations [20, 24].

Incarceration is associated with higher occurrence of infectious diseases, with expressive variation between countries. However, there is a consensus that, when compared to the general population, the prevalence of both HIV and HCV is extremely superior in these contexts [18, 25, 26]. Despite all the importance around the topic, there seems to exist a gap in studying and reporting data relating STIs in prisons and its potential impact in the community [8, 18]. The seroprevalence found in our study of anti-HCV antibodies (2.7%) and HIV (1.6%) was lower in comparison to global estimates. Indeed, a systematic review showed that around 15.1% and 3.8% of prisoners present HCV and HIV infection, respectively [8].

According to the literature, the susceptibility for HIV infection is amplified in individuals with history of STIs, being aggravated in cases with repeated exposure and/or interrupted treatments [27, 28]. One mechanism that increases this susceptibility the fact that some STIs, such as urethritis, damage epithelial surfaces in the genital tract, thus facilitating other

infections, including HIV [29]. Unprotected sex between prisoners can facilitate STIs transmission. These infections, when untreated, can lead to rapid spread within the prison environment and in the external community as well [30]. Our study showed that, when compared to individuals who received intimate visit, those who reported not receiving it had almost nine times more chances of HIV infection. Important for the context of this study, risky sexual relations can occur in exchange for benefits or due to coercion (i.e., sexual violence). Consequently, there is a complex interaction of multiple determinants of risky behaviors among this population.

IDU seems to be the main risk factor associated with HCV infection in prison populations. Fiore and collaborators provided evidence that IDU were significantly less likely of receiving previous treatment for HCV in comparison to inmates without an active infection [6]; indeed, recent studies indicate that IDU have up to eight times more risk of contracting this virus [18, 21, 23], which is very similar to the results reported here. From an epidemiological perspective, these findings–albeit elevated–are not surprising, given that IDU in prisons account for up to 38% of inmates in Europe and 55% in Australia; in these same regions, IDU in the general population is estimated to be around 0.3% [31]. Data from another Brazilian state revealed a prevalence ratio of 5.3% (95% CI 1.97–13.37) of inmates with active HCV infection that were also IDU [24].

Older age and recidivism were risk factors associated with positive anti-HCV serological results in the current study. This could be related to prolonged exposure to biobehavioral variables that underpin STI infection [21]. Notably, these findings are aligned with past reports [32, 33]; however, they could represent some failure by the state in protecting individuals under custody and when they leave the penitentiary system [9]. Indeed, evidence suggests higher syringe sharing, HIV and HCV infection, and increase mortality shortly after inmates are released [34, 35]. Thus, a change in the environment can result in interruptions on a series of protective measures, which once again support the idea of continued assistance to those who enter the penitentiary system [18].

The multivariate model did not maintain the role of homosexual relations and piercing usage–significant in the analyses reported in Table 2 –in relation to positive anti-HCV serological results. In addition, the apparent contradicting result showing that individuals without piercing had a higher seroprevalence of anti-HCV antibodies must be examined carefully. Even if there is scarce data on the prevalence of piercings within prisons, it is undeniable that this practice poses a potential health risk since instruments for skin perforation can be contaminated by infected blood or body fluids, thus increasing the changes of acquiring STIs, such as HCV [36, 37].

Preventive campaigns were related to increased risk for HIV, which again might appear surprising at glance. However, the dynamic of reverse causality could underpin these results. As such, although the inmates have knowledge about STIs, they perhaps do not translate these into protective behaviors. It could also be the case that, for being already infected with HIV, those individuals are perhaps more prone in attending such campaigns for various reasons, such as expanding their social support network or sharing their very own experiences to help others. Indeed, research suggests that more than half of the prison population enter the system without any prior information on STIs and HIV [18, 38].

In summary, the study shows relevant data to public health and contributes with the lack of information concerning STIs in the studied population [19]. The seroprevalence of anti-HCV antibodies (2.7%) is relatively lower when compared to other national and international studies that show estimates ranging from 2.4% to 17% [21, 24, 39, 40]; nonetheless, the presence of anti-HCV antibodies was five times higher in this study than the estimated prevalence (0.53%) for the Brazilian population [41]. In turn, HIV prevalence (1.6%) was lower if compared to the

3.8% estimates for the global prison population [8]; it was likewise below the indices seen among key Brazilian populations, such as sex workers (5%), gays and other men who have sex with men (18%), transgender people, people deprived of liberty and people who use alcohol and other drugs (5%), yet higher when compared to the general population (0.4%) [42].

Failure to detect and promptly treat prisoners' health, as well as the denial of factors that contribute to the spread of STIs result in higher costs for both detainees and taxpayers [38]. Therefore, it is advisable to conduct further studies targeting the prison population to obtain clearer estimates of STIs in the country [43]. Efforts aiming to better understand the dynamics of transmission, including organizational (testing policies, access to medicines, condom supply), pathophysiological (genotypes, resistance, co-infections, acceptability of medications), and violence aspects among inmates (blood-blood contact, sexual contact) might guide STIs programs in prisons. Moreover, treatment with direct-acting antivirals of HCV-infected inmates is paramount for reducing virus circulation, with possible benefits to the general population [44–46].

Finally, readers are adverted that this study has some important limitations, including a relatively low number of individuals with positive serological results, which could interfere in the accurate calculation of OR. Moreover, we performed serologic testing for HCV but were unable to confirm active infections by HCV-RNA, which certainly needs to be accounted in future studies for a more precise estimation of active infections [6]. In addition, findings could not reflect the situation of other types of prisons, including non-closed regimen, juvenile detention centers, and women's prisons [20, 24]. Future studies might take these limitations into account to expand our knowledge on the burden of STIs in the Brazilian prisons.

## Conclusions

Prisons are institutions with intensive concentration of adverse social, economic, and structural factors that facilitate the dissemination of viruses. The fragile infrastructure of Brazilian detention centers, including the lack of proper health care, contribute to prisoners' higher exposure to illnesses, which characterizes this population as key vectors in propagating HIV and HCV. As healthcare services in prisons are severely affected due to numerous factors, a change in perspective is needed since these environments may be crucial for communities in reducing the circulation of STIs. Thus, policies aiming at testing for HIV, HCV, and other infections upon admission and during incarceration are urgent.

Moreover, efforts in understanding risk factors associated with HIV and HCV are paramount to guide robust public policies, grounded on the knowledge of STIs dynamics and their socioeconomic and cultural specificities. Inside these institutions, harm reduction measures have shown encouraging results, albeit external communities still condemn these practices. Consequently, interventions need to take place inside and outside prisons, ensuring that public health services are available after release and that behaviors that maintain stigma and prejudice are tackled within communities.

## Author Contributions

**Conceptualization:** Lirane Elize Defante Ferreto, Stephanny Guedes, Fernando Braz Pauli, Franciele Aní Caovilla Follador, Harnoldo Colares Coelho, Guilherme Welter Wendt.

**Formal analysis:** Lirane Elize Defante Ferreto, Harnoldo Colares Coelho.

**Investigation:** Samyra Soligo Rovani, Franciele Aní Caovilla Follador, Ana Paula Vieira, Renata Himovski Torres, Guilherme Welter Wendt.

**Methodology:** Lirane Elize Defante Ferreto, Stephanny Guedes, Fernando Braz Pauli, Franciele Aní Caovilla Follador, Ana Paula Vieira, Renata Himovski Torres, Harnoldo Colares Coelho, Guilherme Welter Wendt.

**Project administration:** Lirane Elize Defante Ferreto, Renata Himovski Torres.

**Resources:** Renata Himovski Torres.

**Writing – original draft:** Lirane Elize Defante Ferreto, Stephanny Guedes, Fernando Braz Pauli, Samyra Soligo Rovani, Franciele Aní Caovilla Follador, Harnoldo Colares Coelho, Guilherme Welter Wendt.

**Writing – review & editing:** Stephanny Guedes, Samyra Soligo Rovani, Franciele Aní Caovilla Follador, Ana Paula Vieira, Renata Himovski Torres, Guilherme Welter Wendt.

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
