## [Decision Letter · Decision Letter 0]

12 May 2021

PONE-D-21-05227

Prevalence and associated factors of HIV and  Hepatitis C in Brazilian high-security prisons: A state-wide epidemiological study

PLOS ONE

Dear Dr. Wendt,

Thank you for submitting your manuscript to PLOS ONE. After careful consideration, we feel that it has merit but does not fully meet PLOS ONE’s publication criteria as it currently stands. Therefore, we invite you to submit a revised version of the manuscript that addresses the points raised during the review process.

Your manuscript was reviewed by 2 experts in the field. Both identified many important problems in your submission and provided copious comments. Please consider the attached comments and provide point-by-point responses.

We look forward to receiving your revised manuscript.

Kind regards,

Yury E Khudyakov, PhD

Academic Editor

PLOS ONE

Journal Requirements:

2. Please provide additional details regarding participant consent. In the ethics statement in the Methods and online submission information, please ensure that you have specified whether consent was informed.

3. Please provide additional information regarding the considerations  made for the prisoners included in this study. For instance, please discuss whether participants were able to opt out of the study and whether individuals who did not participate receive the same treatment offered to participants.

4. In your Methods section, please provide additional information about the participant recruitment method and the demographic details of your participants. Please ensure you have provided sufficient details to replicate the analyses such as:

a) the recruitment date range (month and year),

b) the names of the 11 maximum security prisons,

c) a table of relevant demographic details,

d) a statement as to whether your sample can be considered representative of a larger population, and

e) a description of how participants were recruited.

5a) If there are ethical or legal restrictions on sharing a de-identified data set, please explain them in detail (e.g., data contain potentially identifying or sensitive patient information) and who has imposed them (e.g., an ethics committee). Please also provide contact information for a data access committee, ethics committee, or other institutional body to which data requests may be sent.

5b) If there are no restrictions, please upload the minimal anonymized data set necessary to replicate your study findings as either Supporting Information files or to a stable, public repository and provide us with the relevant URLs, DOIs, or accession numbers. Please see http://www.bmj.com/content/340/bmj.c181.long for guidelines on how to de-identify and prepare clinical data for publication. For a list of acceptable repositories, please see http://journals.plos.org/plosone/s/data-availability#loc-recommended-repositories.

Reviewers' comments:

Reviewer's Responses to Questions

**Comments to the Author**

1. Is the manuscript technically sound, and do the data support the conclusions?

Reviewer #1: Partly

Reviewer #2: Partly

2. Has the statistical analysis been performed appropriately and rigorously? 

Reviewer #1: Yes

Reviewer #2: Yes

3. Have the authors made all data underlying the findings in their manuscript fully available?

Reviewer #1: No

Reviewer #2: No

4. Is the manuscript presented in an intelligible fashion and written in standard English?

Reviewer #1: Yes

Reviewer #2: Yes

5. Review Comments to the Author

Reviewer #1: Reviewer comments

Manuscript ID: PONE-D-21-05227

Title" Prevalence and associated factors of HIV and Hepatitis C in Brazilian high-security

3 prisons: A state-wide epidemiological study".

Generally speaking:

Thank you for providing me the opportunity to review this manuscript that raises important critical issues about assessing the Prevalence and associated factors of HIV and Hepatitis C in Brazilian high-security 3 prisons. Such assessment will provide concrete data that will help in the micro-elimination of HCV. Definitely, the awareness about what is going on in prisoners is essential and would be great if the authors could provide in addition the situation of the action taken by the government e.g treatment options, strategy development and what is planned for this particular at risk groups

Following are my specific comments:

Comment:1

1. Title: is specific and reflective of the research work content.

Comment:2

2. Abstract:

a) Background –correct with a sentence focused on a key idea explaining why this topic was chosen.

b) Objectives – add the risk factors under the investigation

c) Methodology - correct

d) Results - need some quantification on the risk factors ….

e) Conclusions- are too general; the conclusion should mention specific challenges to be met as well as specific recommendations that is related to the findings of the manuscript.

Comment:3

3. Introduction:

Well written explaining why this topic was chosen for analysis in this article.

Comment:4

Methods:

Generally, the information mentioned under the methodology deficient and should be divided into several sub-sections as follow:

a. Type of the study

b. Study setting

c. Study Participants:

d. Sample type and size: The basis of sample size calculation should be mentioned to know the confidence level and the margin of error.

e. Measurements- OR is calculated for both the comparative cross section study and case control study as well. Please refer to the comment 5-2 in the study section.

Comment:5 Results: has to be badly improved for all the tables

5. 1 Results – should be separated from the discussion section

5.2 The mentioned sample size was 1,154 with overall prevalence of HCV and HIV detected among 1,132 meaning that only 22 cases were –ve. Accordingly, the authors has to mention how they calculated the Odds Ratio with this very small number of the comparative group. This imply that All the tables should include two columns for number and % of HCV +ve and HCV _ve and 2 columns for HIV +ve and –ve for the results to be more understandable.

5.2 line 140 and 141 indicated that the prevalence of HCV (2.7%) and HIV (1.6%) while the tables mentioned an overall prevalence 1,132: please clarify

Comment:6

6. Discussion: should be separate section, focusing discussion on critical or essential findings and explicitly linking the conclusions with the reported data should be more emphasized.

Comment:7

7. Conclusion – is general one, it should be specific and explicitly linking the conclusions with the reported data, write suggestions for improvement as well as add limitations of the proposed technique to the conclusion section and provide recommendations for future research that is focused on the findings

8. References: adequate

Reviewer #2: Having reliable data on the HIV and HCV infection spread within closed prison communities is certainly very important, especially in this era of highly active antivirals against the two viruses. To achieve the goal of reducing reservoirs of infection in humans and the infections spread, until eradication for HCV, it is necessary to identify all difficult-to-reach and unaware patients in the few places where this is possible. Prisons are certainly one of these. For this reason I think this paper have a great interest. Unfortunately, the manuscript present many critical and incomprehensible points, starting with how the sample was collected. The authors speak of a randomization of 8,142 inmates of 11 correctional institutions, but do not explain in any way how they arrived at 1,132 HCV screening tests and, above all, how many inmates were tested for each prison (only a generic "prevalence by municipality" is indicated). Furthermore, those who tested positive seems to referred as infected without an HCV-RNA determination and without having been defined the proportion of viraemic HCVs to be treated with DAA.

For these reasons, we think the paper can be improved and become sufficient for publication on PLOS ONE with the following corrective actions:

- To improve the comprehensibility and clarity of the study, our advice is to separate the chapter of materials and methods into paragraphs as follows: study conduction, sample size and statistical analysis, ethical issues. Please provide sample size determination including it in the described paragraph or mention it as limitation of the study.

- Similarly, it is advisable to separate the results and discussion section into two paragraphs. The results should be described more systematically and subsequently with no judgements on the results. In a separate chapter the discussion should compare the results with international literature in total and stratified in the different cohorts, with special regard on IDU patients cohort. Here a list of examples in literature that may help:

DOI 10.1007/s10654-014-9958-4

- HCV serologic positivity found in enrolled patients if not followed by determination of a positive HCV-RNA cannot be considered as active infection. It is never mentioned if viral load in patients with positive antibodies for HCV was performed. Throughout the text of the article the term HCV infection is therefore used improperly and it should be replaced. We believe for ethical concerns HCV-RNA determination should be at least scheduled in positive HCV antibodies patients to help diagnosis and start treatment. Nevertheless this data could be included in the study and compared. Here a list of other examples in literature:

https://doi.org/10.1111/liv.14745

https://doi.org/10.1016/j.drugpo.2018.06.017

https://doi.org/10.1016/j.drugpo.2020.103055

- There is no mention of the study limitations.

- The conclusions of the study that call for urgent change in perspective are not supported by the evidence of the study itself that shows relatively low HCV seroprevalence.

6. PLOS authors have the option to publish the peer review history of their article (what does this mean?). If published, this will include your full peer review and any attached files.

Reviewer #1: No

Reviewer #2: **Yes: **Babudieri Sergio

---

## [Author Response · Author response to Decision Letter 0]

31 May 2021

To the editorial board:

Many thanks for considering this manuscript for publication. On behalf of all the collaborators, please accept our gratitude in receiving such a constructive feedback on our paper. We broaden this sentiment to the reviewers, who gave us not only their precious time but also their insightful analysis. 

Responses to every comment and suggestions are detailed below. Changes to the revised manuscript are now highlighted in red color. 

Please, do not hesitate to get in contact if there are any other issues involving our article that can be clarified.

Yours sincerely,

Guilherme Welter Wendt, PhD

 EDITORIAL

Comment e.1: Please ensure that your manuscript meets PLOS ONE's style requirements, including those for file naming. The PLOS ONE style templates can be found at https://journals.plos.org/plosone/s/file?id=wjVg/PLOSOne_formatting_sample_main_body.pdf and

Response e.1: The manuscript has been reviewed and updated to adhere more closely to PLOS ONE’s style requirements. We carefully investigated the template, and all the formatting seems to be correct now – including file naming.

Comment e.2: Please provide additional details regarding participant consent. In the ethics statement in the Methods and online submission information, please ensure that you have specified whether consent was informed.

Response e.2: We have provided further details on the processes of informed consent employed by this study. These processes were reviewed and approved by the Human Research Ethics Committee of the Western Paraná State University and adhered with Brazil’s National Statement on Ethical Conduct in Human Research and are highlighted in the revised manuscript (p. 6, under the procedures section).

Comment e.3: Please provide additional information regarding the considerations made for the prisoners included in this study. For instance, please discuss whether participants were able to opt out of the study and whether individuals who did not participate receive the same treatment offered to participants.

Response e.3: We thank the editor for pointing this out and we have clarified this info. The revised version now contains these descriptions (p. 6, section “Procedures”).

Comment e.4: In your Methods section, please provide additional information about the participant recruitment method and the demographic details of your participants. Please ensure you have provided sufficient details to replicate the analyses such as:

a) the recruitment date range (month and year),

b) the names of the 11 maximum security prisons,

c) a table of relevant demographic details,

d) a statement as to whether your sample can be considered representative of a larger population, and

e) a description of how participants were recruited.

Response e.4: All these points were addressed and appear in the revised version of our manuscript. Hence, information on a) is given on the page 5, section Study design and setting; b) appears on the page 5, section Study design and setting; c) now appears in a new table (Table 1); d) is on page 5, also flagged in red color; and e) is on pages 5-6, within the procedures section. 

Comment e.5: We note that you have indicated that data from this study are available upon request. PLOS only allows data to be available upon request if there are legal or ethical restrictions on sharing data publicly. For information on unacceptable data access restrictions, please see http://journals.plos.org/plosone/s/data-availability#loc-unacceptable-data-access-restrictions.

5a) If there are ethical or legal restrictions on sharing a de-identified data set, please explain them in detail (e.g., data contain potentially identifying or sensitive patient information) and who has imposed them (e.g., an ethics committee). Please also provide contact information for a data access committee, ethics committee, or other institutional body to which data requests may be sent.

5b) If there are no restrictions, please upload the minimal anonymized data set necessary to replicate your study findings as either Supporting Information files or to a stable, public repository and provide us with the relevant URLs, DOIs, or accession numbers. Please see http://www.bmj.com/content/340/bmj.c181.long for guidelines on how to de-identify and prepare clinical data for publication. For a list of acceptable repositories, please see http://journals.plos.org/plosone/s/data-availability#loc-recommended-repositories.

Response e.5: In our revised cover letter, we addressed the prompts 5a and 5b. In short, research with human beings is extremely strict in Brazil and sometimes international scholars “get annoyed” with our local guidelines. We would like that the process was simpler, but unfortunately, we must adhere with the law.

The issue of sharing our de-identified data set involves the nature of the sample (defined as vulnerable by the National Health Council - NHC) and the conditions imposed by the institutions/prisons and our State Penitentiary Department (DEPEN-PR), that did not give prior consent for sharing data with others. According to the NHC resolutions (that must be followed by every Research Ethics Committee, including the one which approved our study) there are some impediments after the year of 2012 when sharing certain type of data, thus resulting in ethical and legal impediments.

As such, the Resolution NHC number 466 (http://conselho.saude.gov.br/resolucoes/2012/Reso466.pdf), items “II.25”, “III 1 q”, and “IV.3 e” states that researchers might use and share the material and data obtained in the research exclusively according to the consent of the participant and/or institutions and sharing these details must be approved beforehand. At the time of data collection, we did not ask for participants and institutional approval for sharing data, including the main institution (DEPEN) and its 11 high security prisons. This was entirely our fault as we did not anticipate this request. 

Moreover, NHC has special policies around vulnerability to protect people or groups that, for whatever reasons, have their capacity for self-determination reduced or hindered, or in any way are prevented from opposing resistance, especially with regards to informed consent. For these reasons, most Research Ethics Committee only approve research with this type of sample if scholars assure and limit access to data to a few individuals. 

Consequently, we provided now extra information on how to proceed in case of requesting our data. In summary, a justification must be addressed to every prison unit, and then to the State Penitentiary Department, which would examine the request and provide or deny the sharing to third parties; after that, we would have to amend our ethical application with the NHC and with the Western Paraná State University. This process usually does not take long and those interested in accessing the data would receive all our support. We hope this clarification is helpful. We provided the contact details as requested too in the cover letter.

 REVIEWER 1

Comment 1.1: 

Abstract:

a) Background – correct with a sentence focused on a key idea explaining why this topic was chosen.

b) Objectives – add the risk factors under the investigation

c) Methodology - correct

d) Results - need some quantification on the risk factors ….

e) Conclusions- are too general; the conclusion should mention specific challenges to be met as well as specific recommendations that is related to the findings of the manuscript.

Response 1.1: We thank the reviewer for these comments on the abstract. We have incorporated all the suggestions that are now clearly highlighted in red color. Thus, we added the key idea of why the topic was chosen (scarcity of studies on HCV and HIV among prisoners, mainly in developing countries, p. 2, lines 30-32), the risk factors investigated (p. 2, line 33) and their quantification (p 2., lines 39-44). We also tried to be specific in the conclusion (pp. 2-3, lines 45-51). In summary, the abstract looks much more complete, informative, and convincing after incorporating these suggestions. Any other comments would be very welcomed.

Comment 1.2: Methods:

Generally, the information mentioned under the methodology deficient and should be divided into several sub-sections as follow:

a. Type of the study

b. Study setting

c. Study Participants:

d. Sample type and size: The basis of sample size calculation should be mentioned to know the confidence level and the margin of error.

e. Measurements- OR is calculated for both the comparative cross section study and case control study as well. Please refer to the comment 5-2 in the study section.

Response 1.2: Our thanks go to the reviewer for these suggestions. In the revised manuscript, we have included all the suggested subheadings (pp. 5-7) and expanded on the sample size calculation (p. 6, lines 125-127), allowing readers to know the confidence level and the margin of error.

Comment 1.3: Results: have to be badly improved for all the tables. Should be separated from the discussion section.

Response 1.3: Following the reviewers’ suggestion, we also separated the result section from the discussion. The revised version of the manuscript now includes some important improvements, including a table with demographic data and the extra details requested (i.e., number of + tests, pp. 10-11). We think that is redundant to provide the number of negative tests since it would pollute the table and is basically the total number of people tested (1,132) minus the positive cases indicated in the columns added). All these changes are clearly highlighted in the file with red font. If is in the opinion of the reviewer that further adjustments are necessary, please, do let us know.

Comment 1.4: 5.2 The mentioned sample size was 1,154 with overall prevalence of HCV and HIV detected among 1,132 meaning that only 22 cases were negative. Accordingly, the authors have to mention how they calculated the Odds Ratio with this very small number of the comparative group. This imply that All the tables should include two columns for number and % of HCV +ve and HCV negative and 2 columns for HIV + and HIV – for the results to be more understandable.

5.2 line 140 and 141 indicated that the prevalence of HCV (2.7%) and HIV (1.6%) while the tables mentioned an overall prevalence 1,132: please clarify

Response 1.4: Apologies for this typo in the methods section; the correct sample size is the one informed in the study’s abstract (1,132). We corrected Table 1 (that now reads as Table 2 after adding a table with demographics [new Table 1]) and the information on the prevalence spotted by the reviewer now matches the sample size (p. 6, line 122). Columns were added with positive results. Once again, if needed, we can add the negative tests, albeit we do believe that it is redundant since it is a matter of subtracting the + values out of the 1,132 participants. Regarding the calculation of the odds ratio, the reviewer is right that odds can be inflated/biased with different group sizes. However, we consulted with a biostatistician and we tried to address these issues by using multivariate procedures followed by 95% CI following past research on the matter already published at Plos One (i.e., https://journals.plos.org/plosone/article?id=10.1371/journal.pone.0139487 and https://journals.plos.org/plosone/article?id=10.1371/journal.pone.0169195). In our study’s limitations, we added this as a factor to be considered as well (p. 18, lines 340-342). Suggestions to improve these issues would be again very welcomed.

Comment 1.5: Discussion: should be separate section, focusing discussion on critical or essential findings and explicitly linking the conclusions with the reported data should be more emphasized.

Response 1.5: This revised manuscript now includes several improvements in the discussion, with addition of substantial new literature. Following the recommendations of both reviewers, we discussed a little further the role of IDU in terms of national and international literature (pp. 14-18) while also linking our conclusions with the reported data. The reviewer might notice that we could have made better word choices now, with shorter sentences and more explicit links with the data collected.

Comment 1.6: Conclusion – is general one, it should be specific and explicitly linking the conclusions with the reported data, write suggestions for improvement as well as add limitations of the proposed technique to the conclusion section and provide recommendations for future research that is focused on the findings.

Response 1.6: The reviewer offers a useful suggestion, which we adopted entirely. Consequently, the revised file contains a new conclusion (p. 19, lines 352-361) that is more focused on our data, that suggests improvements for future studies, while our limitations were presented in the previous section (p. 18, lines 340-348).

 REVIEWER 2

Comment 2.1: The authors speak of a randomization of 8,142 inmates of 11 correctional institutions, but do not explain in any way how they arrived at 1,132 HCV screening tests and, above all, how many inmates were tested for each prison (only a generic "prevalence by municipality" is indicated). Furthermore, those who tested positive seems to referred as infected without an HCV-RNA determination and without having been defined the proportion of viraemic HCVs to be treated with DAA. For these reasons, we think the paper can be improved and become sufficient for publication on PLOS ONE.

Response 2.1: We thank the reviewer for these three initial suggestions. Regarding how we arrived at the final sample of 1,132, the revised manuscript now adds this information (p. 6, lines 122-127). Unfortunately, we did not have funds to test the entirely population, so we had to use a randomized sample. The number of inmates tested for each prison is presented on page 8, Table 1. Finally, we rephased the word “infected” for having “anti-HCV antibodies”. We read the reviewers publications on the topic and admire his expertise. So, further suggestions to improve these issues would be very welcomed.

Comment 2.2: To improve the comprehensibility and clarity of the study, our advice is to separate the chapter of materials and methods into paragraphs as follows: study conduction, sample size and statistical analysis, ethical issues. Please provide sample size determination including it in the described paragraph or mention it as limitation of the study.

Response 2.2: We followed all these suggestions. The reviewer will find the separation of the methods section into paragraphs (pp. 5-7) and sample size determination (p. 6, lines 125-127). If there are any other improvements that the reviewer sees as necessary, we would be glad to work on them.

Comment 2.3: It is advisable to separate the results and discussion section into two paragraphs. The results should be described more systematically and subsequently with no judgements on the results. In a separate chapter the discussion should compare the results with international literature in total and stratified in the different cohorts, with special regard on IDU patients cohort. Here a list of examples in literature that may help: DOI 10.1007/s10654-014-9958-4

Response 2.3: The reviewer rightly notes that results and discussion must form different sections. We performed those changes. As for the comparisons with international literature, we expanded the discussions in terms of the general prison population (p. 15, lines 252-261) and also in terms of inject drug users, including data from Brazil (Mato Grosso State) (p. 14, lines 247-251). We thank the reviewer for suggesting the literature, which has been incorporated in our revised file. Apart from these comments, there are new paragraphs in the discussion clearly marked with red font.

Comment 2.4: HCV serologic positivity found in enrolled patients if not followed by determination of a positive HCV-RNA cannot be considered as active infection. It is never mentioned if viral load in patients with positive antibodies for HCV was performed. Throughout the text of the article the term HCV infection is therefore used improperly and it should be replaced. We believe for ethical concerns HCV-RNA determination should be at least scheduled in positive HCV antibodies patients to help diagnosis and start treatment. Nevertheless, this data could be included in the study and compared. Here a list of other examples in literature:

https://doi.org/10.1111/liv.14745

https://doi.org/10.1016/j.drugpo.2018.06.017

https://doi.org/10.1016/j.drugpo.2020.103055.

Response 2.4: Indeed, the study presents seroprevalence and each test was registered in individuals’ medical records when consent was obtained. We apologize for not making that clear beforehand. As such, in our procedures section (p. 6), we included that “Those who agreed to share the results from the serological testing with the institutions medical personnel had their records updated and, if necessary, guidance was offered to health staff.” We also corrected the usage of “infection” or “HCV infected” in the text as already mentioned.

The literature recommended by the reviewer has been extremely useful in strengthening our paper. Consequently, we not only included these papers to support even more some claims made in the introduction, but we also expanded our discussion in line with previous recommendations given by the same reviewer (“the discussion should compare the results with international literature”). Once again, these are marked in red font.

Comment 2.5: There is no mention of the study limitations.

- The conclusions of the study that call for urgent change in perspective are not supported by the evidence of the study itself that shows relatively low HCV seroprevalence.

Response 2.5: We are thankful for these comments. We included several limitations of our study (p. 18, lines 340-349). In respect to our call for “urgent change”, the rationale we used took also into account the role played by prisoners in propagating HIV and HCV. We amended our conclusions to tone down those claims (p. 19). If there are any other suggestions, we would be glad to tackle.

---

## [Decision Letter · Decision Letter 1]

12 Jul 2021

Seroprevalence and associated factors of HIV and  Hepatitis C in Brazilian high-security prisons: A state-wide epidemiological study

PONE-D-21-05227R1

Dear Dr. Wendt,

We’re pleased to inform you that your manuscript has been judged scientifically suitable for publication and will be formally accepted for publication once it meets all outstanding technical requirements.

Kind regards,

Yury E Khudyakov, PhD

Academic Editor

PLOS ONE

Additional Editor Comments (optional):

Reviewers' comments:

Reviewer's Responses to Questions

**Comments to the Author**

1. If the authors have adequately addressed your comments raised in a previous round of review and you feel that this manuscript is now acceptable for publication, you may indicate that here to bypass the “Comments to the Author” section, enter your conflict of interest statement in the “Confidential to Editor” section, and submit your "Accept" recommendation.

Reviewer #1: All comments have been addressed

Reviewer #2: All comments have been addressed

2. Is the manuscript technically sound, and do the data support the conclusions?

Reviewer #1: Yes

Reviewer #2: Yes

3. Has the statistical analysis been performed appropriately and rigorously? 

Reviewer #1: Yes

Reviewer #2: Yes

4. Have the authors made all data underlying the findings in their manuscript fully available?

Reviewer #1: No

Reviewer #2: Yes

5. Is the manuscript presented in an intelligible fashion and written in standard English?

Reviewer #1: Yes

Reviewer #2: Yes

6. Review Comments to the Author

Reviewer #1: (No Response)

Reviewer #2: All comments have been addressed clearly in the corrected version. The text is now well structured with a clear separation of the paragraphs. The sample size is described and the statistics are reported. We hope this study continues with the determination of active infections in order to be able to estimate the active ones and that the infected patients may reach treatment.

7. PLOS authors have the option to publish the peer review history of their article (what does this mean?). If published, this will include your full peer review and any attached files.

Reviewer #1: **Yes: **Prof. Ammal Mokhtar Metwally

Reviewer #2: No

---

## [Editor Report · Acceptance letter]

16 Jul 2021

PONE-D-21-05227R1 

Seroprevalence and associated factors of HIV and  Hepatitis C in Brazilian high-security prisons: A state-wide epidemiological study 

Dear Dr. Welter Wendt:

I'm pleased to inform you that your manuscript has been deemed suitable for publication in PLOS ONE. Congratulations! Your manuscript is now with our production department. 

Kind regards, 

on behalf of

Dr. Yury E Khudyakov 

Academic Editor

PLOS ONE